



# Controls of fluorescent tracer retention by soils and sediments

Marcus Bork[1,2], Jens Lange[2], Markus Graf-Rosenfellner[1], and Friederike Lang[1]

[1]Soil Ecology, Faculty of Environment and Natural Resources, University of Freiburg, Freiburg, Germany
[2]Hydrology, Faculty of Environment and Natural Resources, University of Freiburg, Freiburg, Germany

**Correspondence:** Marcus Bork (marcus.bork@bodenkunde.uni-freiburg.de)

**Abstract.**

Fluorescent dyes like uranine (UR) and sulforhodamine B (SRB) have been used as artificial tracers in hydrological studies for decades. Recently, efforts have been intensified to enable their application to characterise environmental compartments such as soils and derive insights into the fate of pesticides. However, existing knowledge on the controls of sorption of UR and SRB in soils is still incomplete and poorly standardized. For this reason, we aimed to identify the controls of UR and

5 SRB adsorption in soils and to quantify their impact and possible interactions systematically. To reach this goal, we selectively controlled clay, organic matter (OM) and pH within batch experiments and examined the influence of these controls on the adsorption of UR and SRB in soils and sediments.

Tracer adsorption was studied using a sandy sediment with low content of organic matter (OM), a silty loamy topsoil with

10 2.8 %-organic carbon (OC) and a subsoil with 0.6 % − OC. Sorption isotherms were determined by preparing watery solutions containing six different concentrations steps for each tracer and shaking these solutions with the suspended sorbent for 42 h (sorbent-solution ratio of 1:5). Afterwards, the suspension was centrifuged and the tracer fluorescence was measured in the supernatant. The effect of the pH on tracer adsorption was investigated by adjusting the pH of solution/adsorbent suspensions repeatedly during the equilibration to 5.5, 6.5 and 7.5 by adding hydrochloric acid and sodium hydroxide. Additionally, we

examined the impact of OM and clay on tracer adsorption by conducting the batch experiments at pH 7.5 and manipulating the sorbents: OM was removed from top- and subsoil samples by $H_2O_2$-treatment and the clay mineral montmorillonite was added to the sandy sediment to achieve final clay contents of 0.1, 0.5, 1, 2, 2.5, 5 and 10 %.

The linear sorption coefficient $K_d$ and pH showed a negative relationship which was stronger for UR than for SRB. This observation might be explained by an increasing number of negative sorption sites of sorbents and functional groups of both

tracers with increasing pH. The pH-dependent carboxyl- and hydroxyl groups of the UR-molecule might be the reason for the higher extend of this effect for UR. As expected, UR and SRB adsorption increased with increasing clay content, due to more sorption sites related to an increase of the specific surface. The addition of 4 % of the clay mineral montmorillonite sufficed to adsorb nearly 100 % of both tracers. The complex sorption behaviour of UR and SRB in soils and sediments became obvious since OM influenced their sorption in opposite direction: The adsorption of UR increased with increasing OC concentration

while that of SRB decreased.

Our study indicates the high relevance of physico-chemical properties of soils and sediments for the fate of applied tracers. The investigated controls determine if the respective tracer shows more conservative or non-conservative behaviour. Overall,



the reported results will help to optimise the use of fluorescent tracers in terrestrial ecosystems to maximise their potential as a cheap and fast tool to gain insights into the fate of pollutants in soils and sediments.

## 1 Introduction

Fluorescent dyes are used in various applications, such as chemical sensing (Basabe-Desmonts et al., 2007), dye lasers (Li and Psaltis, 2008) or fluorescent labeling of biomolecules (Giepmans et al., 2006; Resch-Genger et al., 2008; Gonçalves, 2009). They are detectable at low concentrations, simple to handle and characterized by low toxicity (Flury and Wai, 2003; Leibundgut et al., 2009). For this reason, fluorescent dyes have been also used as artificial tracers in hydrological studies for decades to investigate transport processes in surface water bodies and groundwater (Omoti and Wild, 1979b; Sabatini and Austin, 1991; Vanderborght et al., 2002; Hillebrand et al., 2015, e.g.). One of the most frequently used fluorescent tracer is xanthene dye uranine (UR) which is used to analyse flow pathways, measure water velocities and determine hydrodynamic dispersion mainly in groundwater (Flury and Wai, 2003). Uranine is subject to photolysis, but in the dark it is mostly assumed to be conservative (Flury and Wai, 2003), meaning that transport is only affected by convection, diffusion, and dispersion. Recently, researchers additionally used non-conservative tracers, such as sulforhodamine B (SRB) in mixture with conservative tracers. This „multi-tracer approach") characterises environmental compartments and predicts the fate of chemically similar substances, such as pesticides (Sabatini and Austin, 1991; Passeport et al., 2010; Lange et al., 2011; Schuetz et al., 2012; Durst et al., 2013; Maillard et al., 2016). Since many different processes like volatilisation, sorption and degradation by light or microorganisms affect the fate of substances in environmental compartments, different tracers are necessary to mimic these processes. The development of this approach results from an increasing need to assess the capacity of natural soils or artificial soil and wetland systems to retain pollutants and prevent water bodies from pollution. Since the detection and quantification of organic micropollutants (e.g. pesticides) is time-consuming and expensive, it is desirable to have faster and cheaper analysis tools.

First multi-tracer tests were conducted by Sabatini and Austin (1991) who used UR and rhodamine WT (RWT) to trace the behaviour of the herbicides atrazine and alachlor in column experiments. They observed that the retention strength of the herbicides was between that of UR and RWT. Promising results in more recent times were reported by Passeport et al. (2010) and by Lange et al. (2011), who found SRB to show similar breakthrough curves like the pesticide isoproturon within drainage outflow and surface outlet of a constructed wetland with a silty clayey soil. In addition, Durst et al. (2013) found similar transport characteristics of UR and isoproturon in a column-experiment with a silty sediment. These studies suggest that it is generally possible to use fluorescent tracers to mimic the transport of organic pollutants, although detailed insights into internal mobility controlling processes are missing so far, especially for soil systems.

Prerequisite for successful multi-tracer experiments in soils and sediments are similar sorption properties of both, the organic pollutant and the respective tracer. This work focuses on the prominent fluorescent tracers UR and SRB. Due to their functional groups (carboxyl group for UR and sulphonic groups for SRB) it is likely that their sorption behaviour will strongly depend on pH, organic matter (OM) and texture of the sorbing matrix. The physico-chemical environment thus dictates a conservative or non-conservative tracer behaviour.





A few sorption studies of fluorescent tracers were carried out using pure organic or inorganic sorbents but the controls of sorption were examined rather by random than by systematically controlled experiments. Smart and Laidlaw (1977) investigated the sorption of several dye tracers, amongst others UR and SRB, on different organic (humus, sawdust) and inorganic (kaolinite, limestone, orthoquartzite) materials. For UR they found moderate adsorption to organic and inorganic materials whereas SRB showed low adsorption to organic materials but higher sorption on mineral surfaces. However, since the pH was not controlled, tracer adsorption to humus could have been enhanced by a pH effect additionally to the presence of OM. Kasnavia et al. (1999) and Sabatini (2000) conducted sorption experiments with UR and SRB at neutral pH on inorganic materials like negatively charged silica ($SiO_2$) and sandstone and positively charged alumina ($Al_2O_3$) and limestone. UR and SRB adsorbed stronger on positively charged surfaces than on negatively charged ones. They also found a strong decrease of UR-sorption and a less strong decrease of SRB-sorption when they increased the pH from 7 to 9. Recently, Dollinger et al. (2017) found that sorption of SRB in soils mainly correlated with clay content while sorption of UR mainly followed content of organic carbon (OC) and pH.

In the past, UR and SRB fluorescence tracer are mainly used in hydrogeological research to identify water flow in saturated zones such as aquifers and karst regions. In theses applications the complex sorption behaviour of UR and SRB on organic matter and clay at different pH-values often plays a limited role. However, in soils and sediments, sorption processes are more relevant and result in a wider range of $K_d$-values (linear sorption coefficients). Use of UR and SRB in surface water bodies and soils therefore requires more knowledge of sorption processes. Since many researchers use $K_d$-values to describe and model linear adsorption processes, considering their controls is crucial. The aim of our study is, therefore, to identify the controls of UR and SRB adsorption in soils, and to systematically quantify their impact and possible interactions.

We investigated the influence of clay, OM, pH and their interactions on the adsorption of UR and SRB in soils and sediments by selectively controlling each in batch experiments. These types of experiments have the main advantage that experimental conditions are exactly controlled. This is an appropriate tool to investigate the influence of single properties independent of transport processes (i.e. preferential flow) and transport-related soil properties (i.e. porosity). To face the challenge of experiments with complex matrices, such as soils and sediments, we not only compared natural substrates but also manipulated them in a controlled way. For the investigation of the influence of OM, we used samples of top- and subsoil with different OM contents and additionally decreased OM by oxidation similar to studies by Kaiser and Zech (2000). For clay, we followed the procedure by Tahir and Marschner (2016) by adding seven different amounts of the clay montmorillonite to a sandy sediment. For pH, we determined $K_d$-values at specific pH-values as done by Boxall et al. (2002) for the sorption of a sulphonamide antibiotic in variously textured soils. Also, to investigate and model the adsorption of fluorescent dye tracers on organic matter, several researchers have used $K_{OC}$-values (e.g. Sabatini and Austin, 1991; Keefe et al., 2004; Durst et al., 2013). However, their use has some limitations (U.S. Environmental Protection Agency, 1999; OECD, 2000). Hence, we additionally evaluated the suitability of $K_{OC}$-values for investigating adsorption of UR and SRB on organic matter.





## 2 Materials and Methods

### 2.1 The substrates

We collected soil samples from a grassland site in Freiburg (Germany) at depths of 5-15 cm (topsoil) and 50-60 cm (subsoil). The samples had similar pH but different OC content (Tab. 2). Furthermore, we used a sandy sediment (quartz sand) purchased

from Kepes Handesgesellschaft mbH Sand und Kies (Freiburg, Germany). In the following, top- and subsoil, and sandy sediment are termed as substrates. For the experiments, stones and roots were removed by hand. Then, all samples were air-dried and passed through a 2 mm sieve. The residual gravimetric water-content was determined as the difference between the sample weight of air-dried sample and the sample weight after 24 h of drying at $105\,^{\circ}$C. Residual water contents ranged from 0.1 % (sediment) to 2.2 % (topsoil) (see Tab. S1). Therefore, we decided to omit the correction of air-dried samples by water

content and used the air-dried soil mass as reference. Texture was determined using sieving, sedimentation and the pipette method (Gee and Or, 2002). The pH was measured at room temperature (approx. $23\,^{\circ}$C $\pm\ 2\,^{\circ}$C) after over-night equilibration of 10 g sample in 25 mL solution (ultrapure water and 0.01 M $CaCl_2$-solution) with a pH-Meter (Metrohm, Germany). Additionally, we analysed the content of dissolved organic carbon (DOC) in centrifuged (2490 g, 1 h, Heracus Megafuge 40 Centrifuge, Thermo Scientific, MA, USA) and filtered (folded filter, 125 mm, $80\,g \cdot m^{-2}$, Munktell Filter AB, Sweden) soil

extracts with a soil-solution ratio of 1:5 after acidification using a TOC-Analyser (multi N/C 2100 S, Analytik Jena, Germany). Centrifugation and filtration of the samples were necessary to remove solid organic matter and mineral particles.

As a measure of OM we determined the OC of the substrates, which was measured with a CNS-analyser (Vario El Cube, Elementar, Germany) after grinding the samples with a vibratory disc mill (Siebtechnik GmbH, Germany). Since the sediment contained carbonate-bound carbon (1.13 %), its OC content was determined as the difference of the total carbon content before

20 and after heating the sample at $550\,^{\circ}$C. At this temperature, it is assumed that all OC had transformed into $CO_2$ (Bisutti et al., 2004).

The specific surface area (SSA) was determined using $N_2$- adsorption method with a Quantachrome Autosorb-1 instrument (Quantachrome Instruments, FL, USA). Samples were degassed at $80\,^{\circ}$C and the SSA was calculated using the Brunauer– Emmett– Teller (BET) equation by Brunauer et al. (1938).

Amorphous Fe-, Al- and Mn-oxides ($Fe_o$, $Al_o$ and $Mn_o$) were extracted by oxalate according to Schwertmann (1964) and crystalline and amorphous Fe-, Al- and Mn-oxides ($Fe_d$, $Al_d$ and $Mn_d$) by dithionite according to Mehra and Jackson (1958) in order to estimate their contribution to tracer sorption. The molar concentrations of amorphous and crystalline metals equal the sum of oxalate-extracted metals ($Me_o$) and dithionite-extracted metals ($Me_d$), respectively. In addition, we calculated the $Me_o$/$Me_d$-ratio which indicates the fraction of more amorphous metal oxides in relation to the more crystalline oxides. All

element concentrations were determined by ICP-OES (Spectro Ciros CCD, Spectro Analytical Instruments GmbH, Germany).

### 2.2 The tracers

Tracer solutions of UR (Simon & Werner, Flörsheim, Germany) and SRB (Chroma, Münster, Germany) were dissolved in 0.01 M $CaCl_2$ and stored at approximately $6\,^{\circ}$C. To prevent photolytic decay, the amber-glass bottles were wrapped with





aluminium foil. The tracer fluorescence was measured in a synchronous scan method ($\Delta\lambda = 25$ nm, wavelength range: 250 - 650 nm) at maxima of 488 nm (UR) and 560 nm (SRB) using the luminescence spectrometer LS-50B (Perkin Elmer, MA, USA). To ensure 100 % fluorescence intensity for both tracers during measurement, the pH was raised to 9-10 using one drop of 1.5 M EDTA solution. This step was necessary due to the pH-dependency of tracer fluorescence (Smart and Laidlaw, 1977)

(Tab. 3). The calibration of the fluorescence measurements was conducted in watery electrolyte extracts of each substrate in order to exclude matrix effects. For this purpose, 20 g substrate material was dispersed in 100 mL 0.01 M $CaCl_2$ and was agitated overnight using a rotating shaker (3040 GFL, Burgwedel, Germany). After centrifuging at 2490 g for 1 h and filtration by a folded filter, the extracts were diluted with ultrapure water by a factor of 10 to reduce the background fluorescence. The calibration solutions were prepared in the range of 0 - 5 $\mu g \cdot L^{-1}$ (UR) and 0 - 100 $\mu g \cdot L^{-1}$ (SRB). These concentrations were

chosen to ensure a linear calibration range.

## 2.3  Batch experiments

### 2.3.1  Substrate treatments

The initial pH-values of all samples were adjusted to 7.5, 6.5 and 5.5 using HCl and NaOH in variable concentrations (Fig. 1). At pH 5.5 and 6.5 only the first and the sixth concentration were measured in triplicate, while the other four concentrations in

between were single measurements. The small variance of the triplicate measurements (6.2 % ± 4.5 %, see Fig. S1) made this simplification possible. The effect of varying OC and clay contents was studied at pH 7.5. In the following, altered substrates, either by $H_2O_2$-treatment (low and high OC) in the case of top- and subsoil or by the addition of clay (Clay0, Clay1 and Clay2) in the case of sediment or by adjusting the pH are termed as treatments.

To decrease OC concentration, approximately 200 g of each soil sample (top- and subsoil) was dispersed in a $H_2O_2$ solution

(w = 30 %) and equilibrated at 20 °C four 15 h. Afterwards, the suspension was heated for approximately two weeks at 80 °C until evolvement of $CO_2$ ended. Within this time, $H_2O_2$ was repeatedly added to the suspension. Subsequently, the sample was rinsed with ultrapure water on a folded filter and dried at 40 °C. By this treatment the OC content decreased from 2.8 to 0.7 % OC (topsoil) and from 0.6 to 0.1 % OC (subsoil) (Tab. 2).

The clay content was modified by adding different quantities of the clay mineral montmorillonite (K10, powder, Sigma

Aldrich, St. Louis, MO) to the sediment. To investigate the influence of the clay content on the tracer sorption for a broader range of clay contents, we first conducted single adsorption measurements at different quantities of clay addition (0.1 %, 0.5 %, 1.0 %, 2.0 %, 2.5 %, 5 % and 10 % clay). In a second step, we chose three clay contents and used the following treatments for analysing sorption isotherms: Clay0 (sediment, 0.1 % clay), Clay1 (sediment + 1 % clay) and Clay2 (sediment + 2 % clay).

### 2.3.2  Sorption isotherms

Sorption of fluorescent tracers was investigated using the batch equilibrium method according to the OECD guideline 106 (OECD, 2000). For each substrate, 10 g was suspended in 45 mL 0.01 M $CaCl_2$ solution in 100 mL amber-glass bottles and agitated overnight in a rotating shaker (3040 GFL, Burgwedel, Germany) at 10 rpm. Subsequently, 5 mL of the tracer solution





was added to the suspension and the pH was adjusted to 5.5, 6.5 or 7.5 by adding HCl or NaOH. The tracer solution contained both tracers at the same. We assumed that the tracers did not interact in solution and did not compete during sorption due to an excess of sorption places and low tracer concentrations (linear sorption range). In the following, the tracer-soil suspension was agitated for 42 h until sorption equilibrium. This time was identified in pre-tests where we tested the tracer adsorption

by conducting the batch experiment for 24 h, 48 h and 72 h. During the equilibration time, the pH was measured two times (after approx. 16 - 24 h) and was, if necessary, corrected again by adding HCl or NaOH. After equilibration, solid and liquid phase were separated by centrifugation at 2490 g for 45 min. Following this, 5 mL of the supernatant was separated and diluted by 45 mL ultrapure water in order to suppress DOC-fluorescence and therefore increase the limit of quantification (LOQ) of tracers. For each sorption isotherm six tracer concentrations in ranges of 20 - 45 $\mu g \cdot L^{-1}$ (UR) and 400 - 900 $\mu g \cdot L^{-1}$ (SRB)

were prepared in triplicate. In pre-tests these concentrations were found to be high enough to obtain a clear tracer signal to background ratio and to be low enough to ensure linear sorption isotherms. Moreover, the experiments were conducted at room temperature (approx. 23 °C $\pm$ 2 °C); Fernández-Pascual et al. (2018) showed that adsorption of UR and SRB is not temperature dependent at this temperature range.

### 2.3.3 Data analysis

The content of adsorbed tracer q (eq) [$\mu g \cdot kg^{-1}$] at chemical equilibrium was calculated as follows:

$$q(eq) = \frac{(c_0 - c_{aq}(eq)) \cdot V_0}{m_{sample}} \tag{1}$$

where $c_0$ [$\mu g \cdot L^{-1}$] is the initial tracer concentration, $c_{aq}$ (eq) [$\mu g \cdot L^{-1}$] is the measured tracer concentration in the liquid phase at chemical equilibrium, $V_0$ [L] is the sample volume and $m_{sample}$ [g] is the sample weight. Since sorption data were still in the linear range, the adsorption percentage $x_{adsorbed}$ [%] was calculated according to the equation:

$$x_{adsorbed} = \frac{m_{sorb}}{m_0} \cdot 100 \ (\%) \tag{2}$$

with $m_{sorb}$ [$\mu g$] = q(eq) * $m_{sample}$ the mass of adsorbed tracer on the soil and $m_0$ [$\mu g$], the initial mass of tracer. The linear sorption coefficient $K_d$ [$L \cdot kg_{soil}^{-1}$] is defined as follows:

$$K_d = \frac{q(eq)}{c_{aq}}. \tag{3}$$

It is determined as slope of the linear sorption isotherms and a measure of sorption strength. We decided to use this equation

because it is widely used and enables comparison with literature data. To evaluate the goodness of $K_d$-values, standard errors and 95 %-confidence intervals were calculated. Linear regression was performed using the lm-function of the R programming

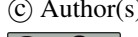



language (R Development Core Team, 2008). Relating the $K_d$ to the fraction of organic matter ($f_{OC}$, [$kg_C \cdot kg_{soil}^{-1}$]) yields the $K_{OC}$ [$L \cdot kg_C^{-1}$], a measure for adsorption on organic matter::

$$K_{OC} = \frac{K_d}{f_{OC}}. \tag{4}$$

Data analysis was performed using the R programming language (R Development Core Team, 2008).

## 3 Results and discussion

### 3.1 Physico-chemical properties of substrates and treatments

The three substrates topsoil, subsoil and sediment were chosen to represent some common occurring natural materials. In agreement, the OC content of the topsoil (2.7 %, Tab. 2) was in accordance with that of other terrestrial European topsoils (de Brogniez et al., 2015). In contrast to the topsoil, the subsoil showed a low OC content (0.6 %) which is related to the relatively low depth of the taken sample (50-60 cm). This was intended to have samples with similar texture but a preferable different OC content. In the sediment we measured a slightly higher OC content (0.8 %) than in the subsoil. However, this value has to be considered carefully because the high inorganic carbon content of the sediment could have influenced the accuracy of the measurement of organic carbon. In contrast to OC, the clay-contents of top- and subsoil are similar (24.0 and 22.4 %) and differ clearly from that of the sediment (0.1 %) which was intended to examine the influence of the clay on the adsorption of the tracers. In general, top- and subsoil are comparable to those used for example Dollinger et al. (2017) for examining sorption of UR and SRB whereas the sediment is similar to the substrates used by Sabatini (2000).

In general, metal oxide and hydroxide contents of our samples, were comparable to soils in central Germany (Wiseman and Puttmann, 2005). They increased in the following order: sediment « topsoil < subsoil. The $Me_d$ (amorphous and crystalline Fe-, Al- and Mn-oxides) was higher in the subsoil than in the topsoil, indicating more positive sorption places in the subsoil. The $Me_o/Me_d$-ratio was nearly twice as high in the topsoil (0.67) than in the subsoil (0.38) indicating more amorphous metal oxides in the topsoil. Since iron oxides like ferrihydrite and goethite are positively charged at high pH-values (Cornell and Schwertmann, 2003), they are important sorbents for molecules with negative charged functional groups like UR and SRB.

The SSA was twice as high in the subsoil ($24\,m^2 \cdot g^{-1}$) as in the topsoil (12 and $10\,m^2 \cdot g^{-1}$) despite a slightly higher clay content in the topsoil (24.0 %) than in the subsoil (22.4 %). This could be related to a different composition of the clay fraction in both soils or by clogging of clay mineral interlayers by organic matter, especially in the topsoil, which was also observed e.g. by Mikutta et al. (2006b) and Mikutta et al. (2006c). Furthermore, the SSA slightly increased after the $H_2O_2$-treatment, which was also observed by Feller et al. (1992). They explained this observation as follows: the organic matter (OM) partially sticks soil particles together or clocks soil pores, which, after the removal of OM, become accessible again for $N_2$ during the measurement. The lowest SSA was, as expected, measured for the sediment ($1.1\,m^2 \cdot g^{-1}$). By the addition of the pure clay mineral montmorillonite with a high SSA ($249\,m^2 \cdot g^{-1}$) the clay content of the treatments Clay1 and Clay2 was increased to $3.6\,m^2 \cdot g^{-1}$ and $6.1\,m^2 \cdot g^{-1}$.



### 3.2 Influence of pH

#### 3.2.1 Sorption of UR

The adsorption of UR in top- and subsoil strongly decreased with increasing pH (Fig. 2, Fig. 3; data can be found in the supplement: Tab. S2, S3 and S4). The $K_d$-value decreased from pH 5.5 to 7.5 by a factor of 22 in the topsoil and 12 in the

subsoil (Tab. 4). Similar observations concerning the direction of the relation between pH and UR-adsorption were made by other researchers, e.g. by Smart and Laidlaw (1977) and Peterson (2010). Omoti and Wild (1979a) reported a $K_d$-value of $10.3 \, L \cdot kg^{-1}$ at pH 6.5 - 7.0 for UR adsorption in batch-experiment (Tab. 1), which is approximately between the values that we measured ($3.3 \pm 0.4 \, L \cdot kg^{-1}$ at pH 7.5 and $46.7 \pm 7.7 \, L \cdot kg^{-1}$ at pH 6.5). At high pH-values, most solid surfaces in soils (variable charges of functional groups of OM and edges of clay minerals as well as permanent charges of clay minerals) are

negatively charged and become, with decreasing pH, neutral or, at more acidic conditions, positively charged (Blume et al., 2016). The same is the case for the carboxyl and hydroxyl groups of the UR molecule that is determined by its $pK_a$-values (Zanker and Peter, 1958). Therefore, attracting forces between UR and various sorbents at low pH turn into repulsive forces between negative charges at higher pH and, thus, a reduced affinity to adsorption.

At pH 5.5 and 6.5, the adsorption of UR was much higher in the topsoil than in subsoil (Fig. 3). This difference vanished at

pH 7.5. Assuming that the OM content is the main difference between top- and subsoil, we can conclude that the influence of OM on the adsorption of UR is higher in more acidic soils. An explanation could be the following: At pH 7.5 most variable charged sorption places in the soil are negatively charged and the adsorption of UR depends only slightly on the content of OM. In contrast, a large part of the variable charged sorption places are protonated and available for UR-adsorption at pH 5.5. In this case, UR adsorption distinctly depends on the OM content.

In our experiments, we observed much higher $K_d$-values (max. $73.2 \pm 12.8 \, L \cdot kg^{-1}$, Tab. 4) than other researchers reported before (max. approximately $10 \, L \cdot kg^{-1}$ measured e.g. by Sabatini (2000) or Dollinger et al. (2017)). The reason is that, in the past, UR was mainly used as a conservative tracer, thus under conditions where UR did not showed adsorption (high pH, sandy texture). However, if we use UR in a multi-tracer approach to mimic the (sorption) behaviour of similar substances we could use its sorption properties.

#### 3.2.2 Sorption of SRB

In contrast to UR, the adsorption of SRB in top- and subsoil decreased only slightly with increasing pH (Fig. 2, Fig. 3). This observation can be explained by its two sulphonic groups that are negatively charged at pH-values above 1.5 and one positive charge (Kasnavia et al., 1999). Therefore, the charge of SRB does not change over a wide range of pH-values. Furthermore, Polat et al. (2011) have shown that SRB is an amphiphilic molecule, which means that the molecule possesses both hydrophilic

and hydrophobic properties. Consequently, SRB has a complex sorption behaviour and can be attracted by a wide range of surfaces including hydrophobic and negative or positive charged surfaces. This results in low pH-dependence of SRB-adsorption.





### 3.2.3 Comparison of sorption of UR and SRB

The $K_d$-value declined linearly with increasing pH (Fig. 3). This effect was more pronounced for UR than for SRB indicating a stronger pH-dependence of UR adsorption. We explain this observation by different functional groups of UR and SRB. The sulphonic groups of SRB have the same negative charge over a wide range of pH due to its low $pK_a$-value ($pK_a < 1.5$,

Kasnavia et al., 1999) while the carboxyl and hydroxyl groups of UR become more negatively charged with increasing pH due to deprotonation (Zanker and Peter, 1958). Therefore, we assume that UR adsorption increasingly happens through hydrophobic interactions with OM when the pH decreases, but not for SRB because its functional group remains negative charged. This observation could be a hint that the variations in molecular charge of the sorbant with changing pH, is much more important than that of the adsorbing surfaces in the soil.

## 3.3 Influence of OC content

### 3.3.1 Sorption of UR

In both subsoil treatments and the untreated topsoil, approx. 35 % of UR was adsorbed, which corresponded to $K_d$-values of $2.3 \pm 0.5$ - $3.3 \pm 0.4 \, \mathrm{L \cdot kg^{-1}}$ (Fig. 4 and Fig.5; data can be found in the supplement: Tab. S2, S3 and S4). These $K_d$-values are in accordance with those of other researchers who worked with similar substrates at comparable conditions (Tab. 1). For

example, Dollinger et al. (2017) calculated $K_d$-values of 1 - 5 $\mathrm{L \cdot kg^{-1}}$ for UR adsorption in soil samples with 19 - 29 % clay, 1 - 5 % OC at pH above 8.0. Moreover, these results also suggest a low dependence of UR adsorption on OM at pH 7.5. Hence, we suppose that not the number of sorption sites is controlling the UR adsorption but the electrochemical repulsion of UR by negative charged sorbents.

Nevertheless, despite mainly negative sorption sites almost one third of UR was adsorbed at pH 7.5. At these conditions,

adsorption potentially took place by cation bridges (e.g. by $Ca^{2+}$- ions) between the negative functional groups of OM and the UR-molecule (Guggenberger and Kaiser, 2003; Schaumann and Thiele-Bruhn, 2011) or took place on positively charged metal oxides. More or less organic sorption sites are thus not crucially influencing the sorption behaviour of UR. At first sight, these observations contradicts findings of other researchers. Dollinger et al. (2017) found that adsorption of UR mainly depends on OC content and pH and also Smart and Laidlaw (1977) found increasing UR adsorption with increasing humus concentration.

However, as both studies did not control the pH, their findings may have been influenced by low pH rather than by increased OC content.

In the $H_2O_2$-treated topsoil UR adsorption was three times lower than in the other treatments ($K_d$-value of $0.6 \pm 0.3 \, \mathrm{L \cdot kg^{-1}}$, Fig. 5). This result might be attributed to sample treatment. $H_2O_2$ did not oxidise the entire OC to $CO_2$, but produced a number of smaller organic molecules (Mikutta et al., 2005) that remained in the soil pores as indicated by higher DOC concentrations

(Tab. 2). These small organic molecules competed with UR for sorption sites and reduced overall UR adsorption.



### 3.3.2 Sorption of SRB

In contrast to UR, $K_d$-values of SRB decreased monotonically with increasing OC content from $16.4 \pm 1.4 \, \text{L} \cdot \text{kg}^{-1}$ at $0.1\% - \text{OC}$ to $2.8 \pm 0.3 \, \text{L} \cdot \text{kg}^{-1}$ at $2.7\% - \text{OC}$ (Fig. 4 and Fig.5), which corresponded to a 50% reduction of adsorption from $83\%$ to $40\%$. Only at the highest OC content were SRB and UR adsorption similar. Furthermore, we measured stronger SRB adsorp-
tion in the subsoil than in the topsoil at all pH-values (Fig. 2) as was also observed by Vanderborght et al. (2002). According to this result, we assume that SRB adsorption mainly took place on the mineral phase, e.g. on metal oxides and hydroxides that are positive charged due to their high point of zero charge (PZC) (Cornell and Schwertmann, 2003). Coating of mineral surfaces by organic substances reduced mineral sorption places and therefore positively charged sorption sites that led to a reduced adsorption of SRB. This result is in accordance with findings of Smart and Laidlaw (1977) who observed lower SRB
adsorption on organic than mineral surfaces. Furthermore, the results from our batch experiment are also in accordance with results from column experiments in Durst et al. (2013). They used a silty sediment with an high OC content of $5.2\%$ at pH 7.5 and calculated also a relatively low $K_d$-value of $3.2 \, \text{L} \cdot \text{kg}^{-1}$ for SRB adsorption (Tab. 1). However, decreasing adsorption of SRB with increasing OC content has not been reported before.

## 3.4 Influence of clay content

### 3.4.1 Sorption of UR

UR adsorption in the sediment samples without clay addition (Clay0) was minimal (Fig. 4 and Fig. 6; data can be found in the supplement: Tab. S2, S3 and S4). This minimal adsorption tendency of UR at low clay content (and simultaneous low OC-, oxide- and hydroxide-content) was likely caused by the lack of sorption places and by repulsive forces due to negative charges of the sand particles and the UR-molecule. This has been observed by many other researchers (e.g. Smart and Laidlaw, 1977;
Kasnavia et al., 1999; Sabatini, 2000) and is the reason why UR is generally assumed to be a conservative tracer in groundwater studies (e.g. Leibundgut et al., 2009).

However, adsorption of UR increased linearly with clay content ($UR_{ads.} = 38.79 \cdot Clay - 9.33$, $r^2 = 0.988$) and reached $100\%$ adsorbed tracer until $2.5\%$ clay addition (Fig. 6). In general, more clay leads to higher adsorption because of increases in SSA. In our samples the SSA increased from $1.1 \, \text{m}^2 \cdot \text{g}^{-1}$ in treatment Clay0 to $3.6 \, \text{m}^2 \cdot \text{g}^{-1}$ and nearly doubled
$(6.1 \, \text{m}^2 \cdot \text{g}^{-1})$ in treatment Clay2 (Tab. 2).

Not only the clay quantity and the SSA influenced adsorption of UR in our experiments, but other effects may play a role. That became apparent in the comparison between $K_d$-values of the untreated subsoil (high OC) and that of the sediment with $2\%$ clay addition (Clay2). Although the untreated subsoil had 10 times the clay content ($22.4\%$) and 4 times the SSA ($24 \, \text{m}^2 \cdot \text{g}^{-1}$) than the sediment, its $K_d$-value was only half ($2.3 \pm 0.2 \, \text{L} \cdot \text{kg}^{-1}$, Tab. 4). This suggests that the type and
composition of the clay fraction influenced UR adsorption. In our batch experiments, we used the clay mineral montmorillonite with a very high specific surface ($249 \, \text{m}^2 \cdot \text{g}^{-1}$). The soil contained a mix of clay minerals that probably had less total or less accessible sorption places than the fresh montmorillonite. The pores of the topsoil clay fraction could be clogged by organic matter (Mikutta et al., 2006b, c) and, thus, be less accessible than those of the fresh clay mineral montmorillonite.





### 3.4.2 Sorption of SRB

SRB only slightly adsorbed to the sediment without clay addition (Clay0, Fig. 4 and Fig. 6) due to the lack of sorption places. With increasing clay addition, the fraction of adsorbed SRB increased exponentially ($SRB_{ads.} = 97.65 \cdot (1 - e^{(-1.06 \cdot Clay)})$) and reached nearly $100\,\%$ at three percent clay content. This result is in accordance with Dollinger et al. (2017) who concluded

that SRB adsorption is strongly correlated with clay content. In contrast to UR, the adsorption of SRB in relation to clay content increased faster, which potentially could be explained by its higher affinity for mineral surfaces as also observed by Smart and Laidlaw (1977).

    As with UR, the quality of clay affected the adsorption of SRB. When SRB adsorption in the treatments Clay2 and topsoil compared to adsorption in treatments with low OC at pH 7.5 (Fig. 5 and Fig. 6), both treatments show similar SRB adsorption

(approx. $80\,\%$), despite the clay contents differing by a factor of 10 (Tab. 2). A different composition of the clay fraction of the topsoil in comparison to the fresh montmorillonite can also explain these results.

### 3.5   Limitations of $K_{OC}$-values

The most import assumptions for the calculation of $K_{OC}$-values are (i) the main sorption of the target substance occurs on OM and (ii) adsorption mainly occurs by nonpolar interactions (OECD, 2000). Hence, the sorption mechanism should mainly be

simple partitioning between the liquid and the organic phase. The U.S. Environmental Protection Agency (1999) listed further limitations of the $K_{OC}$-concept including an organic fraction ($f_{OC}$) between 1 - 20 percent and a content of swelling clays like montmorillonite has to be low. Due to their $K_{OW}$-values (Tab. 3) and their charged functional groups, UR and SRB can be regarded as more polar tracers. This was also demonstrated by this study and by Kasnavia et al. (1999) who calculated $K_{OW}$-values, found a strong pH-dependence for both tracers, and reported a high hydrophobicity for UR. Therefore, we expect

polar sorption interactions like electrochemical interactions and ligand exchange and also adsorption on mineral surfaces rather than partitioning between liquid and organic phase. A calculation of $K_{OC}$-values from our data would thus be inappropriate. For example, increasing $K_{OC}$-values for both tracers would have been calculated for the sediment treatments with rising clay content, although the content of OM - and adsorption to OM - did not change. The adsorption of the tracers was instead related to more abundant adsorbing mineral surfaces of the clay mineral montmorillonite. Moreover, a decreasing $K_{OC}$-value with

rising pH can be expected due to increasing repulsive forces between negative functional groups of tracers and sorbents. In both cases, a parameter other than the OM content would lead to different $K_{OC}$-values. Hence, $K_{OC}$-values should be regarded as inappropriate parameters to describe or compare adsorption characteristics of UR and SRB.

## 4   Conclusions

Overall, our systematic batch experiments indicated that increasing clay content led to a strong increase of adsorption of both

UR and SRB. In contrast higher OM contents mainly caused lower adsorption of SRB and had only small effect on UR (Fig. 7). Increasing pH led to increasing adsorption of both tracers. Thus, we conclude that (i) adsorption of UR is mainly influenced





by pH, while (ii) adsorption of SRB is mainly influenced by the quality of the substrate: positively by clay and negatively by OM. Furthermore, we conclude that (iii) SRB has a higher affinity to mineral surfaces and (iv) UR has a higher affinity to organic surfaces, especially at low pH-values. Moreover, $K_{OC}$-values should not be used to characterise the sorption of polar fluorescent tracers.

From our experiments no general assumptions can be derived regarding the comparative sorption strength or mobility of the tracers in a given soil. Rather, the conditions under which they are used have to be specified. Only then a statement like "SRB is more sorptive than UR" has validity. Due to the complex adsorption behaviour of UR and SRB in soils and sediments, it will be difficult to find a suitable tracer to completely mimic the sorption characteristics of a specific organic pollutant. Nevertheless, this approach is promising if organic pollutants are classified into larger groups with similar properties and if

tracers are identified representing the sorption and transport of these groups. The potential benefits of such a cheaper, faster and easier to operate approach for first approximations of the environmental fate of organic pollutants make further experiments worthwhile. Our analyses stress the substrate dependence of tracer substances and help standardise procedures to characterise and evaluate the tracer capacities of fluorescent substances for the use in specific substrates.

*Author contributions.* Writing - Orignial Draft: M.B.; Writing - Review & Editing: M.B., J.L., M.G-R. and F.L.; Conceptualization and

Methodology: M.B., J.L., M.G-R. and F.L.; Formal Analysis, Investigation, Validation and Visualization: M.B.; Funding Acquisition: J.L., M.G-R. and F.L.; Project Administration: J.L.

*Competing interests.* The authors declare that they have no conflict of interest.

*Acknowledgements.* We thank Lars Best and especially Stefanie Bittner for their extraordinary support during the extensive laboratory analysis. Furthermore, we thank Kenton Stutz for his detailed and very helpful linguistic and technical corrections of the manuscript. This

research was funded by the Federal Ministry of Education and Research (BMBF) (02WRM1366B) support measure "Regional water resource management (ReWaM)" in the project MUTReWa (Measures for a sustainable approach to pesticides and their transformation products in the regional water management). The article processing charge was funded by the German Research Foundation (DFG) and the University of Freiburg in the funding programme Open Access Publishing.



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





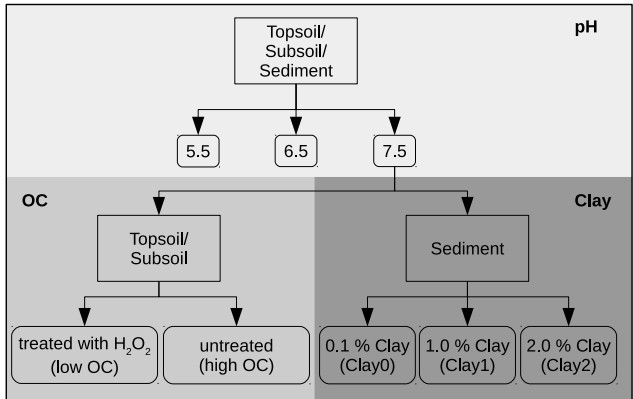

**Figure 1.** Experimental design for the different treatments of the batch experiments.

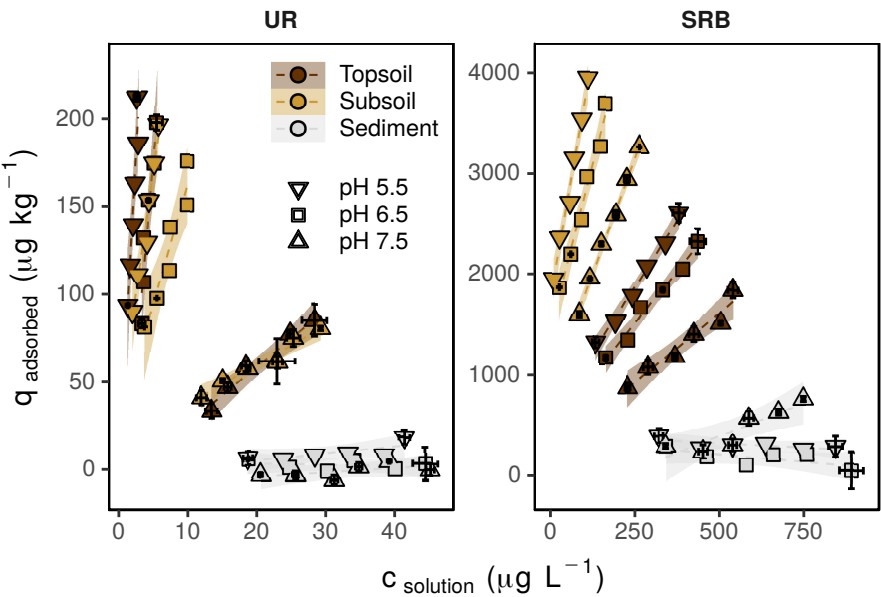

**Figure 2.** Sorption-isotherms for UR and SRB determined at pH 5.5 (down-pointing triangle), 6.5 (square) and 7.5 (up-pointing triangle) in topsoil (brown), subsoil (yellow) and sediment (grey). At pH 5.5 and 6.5 the highest and the lowest concentration of the sorption isotherms were prepared in triplicate, the other concentrations were only prepared once. At pH 7.5 all concentrations were prepared in triplicate. The errorbars in x- and y-direction represent the standard deviation of the tracer concentration measured in solution ($c_{solution}$) and the calculated content of adsorbed tracer ($q_{adsorbed}$) of three replicates each. Note the different concentration ranges of UR and SRB. The transparent areas around the regression lines represent the 95 %-confidence intervals.





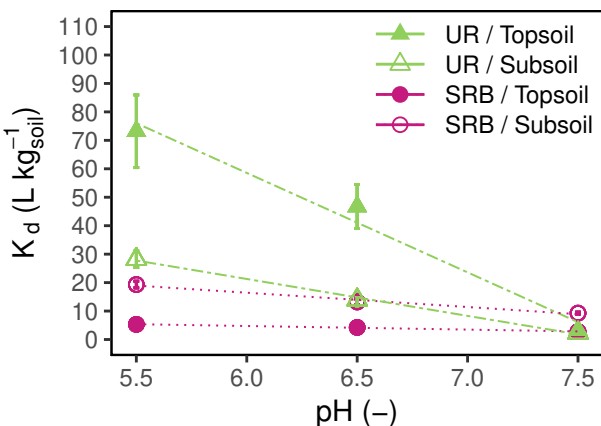

**Figure 3.** $K_d$-values as function of pH for UR (triangles, dash-dotted lines) and SRB (circles, dotted lines) in topsoil (filled symbols) and subsoil (open symbols). Each $K_d$-value was derived as the slope from linear regression of the sorption isotherms (six concentrations times three replications). The error bars represent the standard errors of $K_d$-values from linear regression.



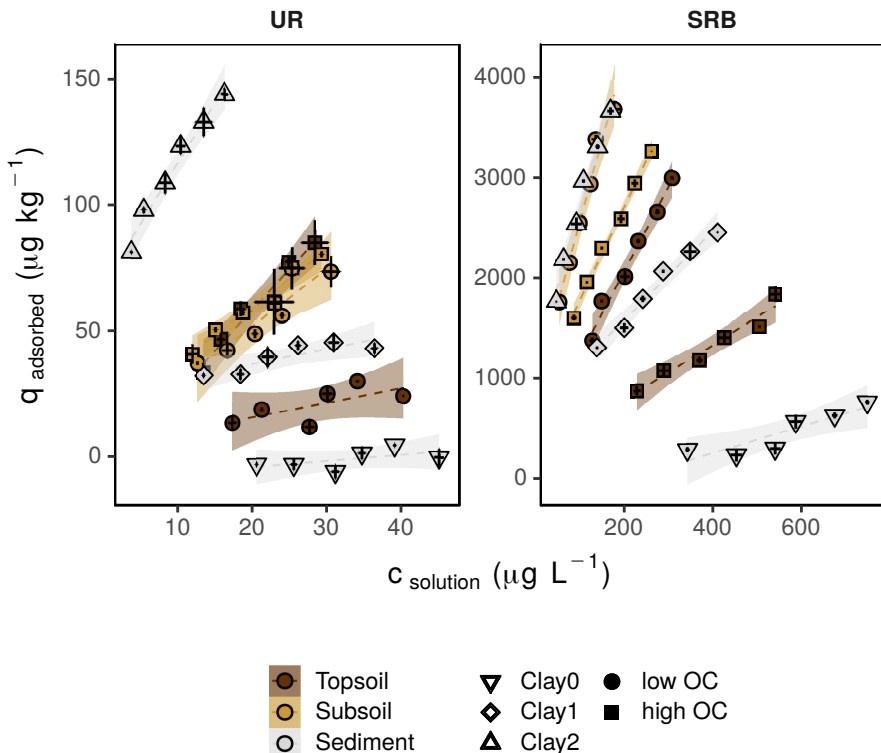

**Figure 4.** Sorption-isotherms for UR and SRB in topsoil (brown), subsoil (yellow) and sediment (grey) at different OC (filled symbols) and clay (open symbols) conditions. The high OC treatments (square) represent the untreated samples, low OC (circle) the $H_2O_2$-treated samples. Clay0 (down-pointed triangle) represents the sediment without clay addition, Clay1 (diamond) and Clay2 (up-pointed triangle) are sediment plus 1 % and 2 % clay addition. All samples were prepared in triplicate. The errorbars in x- and y-direction represent the standard deviation of the tracer concentration measured in solution ($c_{solution}$) and the calculated content of adsorbed tracer ($q_{adsorbed}$) of three replicates each. Note the different concentration ranges of UR and SRB. The transparent areas around the regression lines represent the 95 %-confidence intervals.



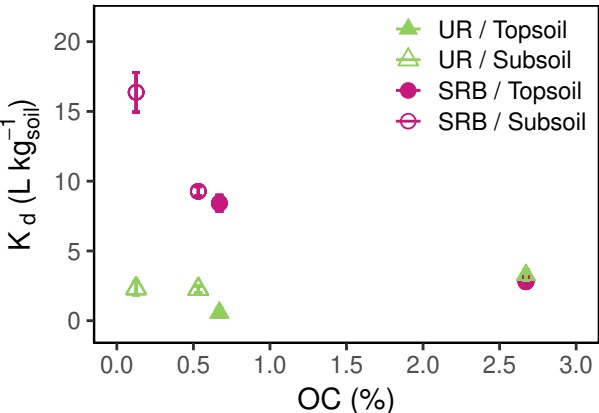

**Figure 5.** $K_d$-values as function of OC (%) for UR (triangles) and SRB (circles) in topsoil (filled symbols) and subsoil (open symbols). For each tracer and type of substrate (top- or subsoil) the $K_d$-value at lower OC-value result from the $H_2O_2$-treatment and the $K_d$ at higher OC from the untreated samples. Each $K_d$-value was derived as the slope from linear regression of the sorption isotherms (six concentrations times three replications). The error bars represent the standard errors of $K_d$-values from linear regression.

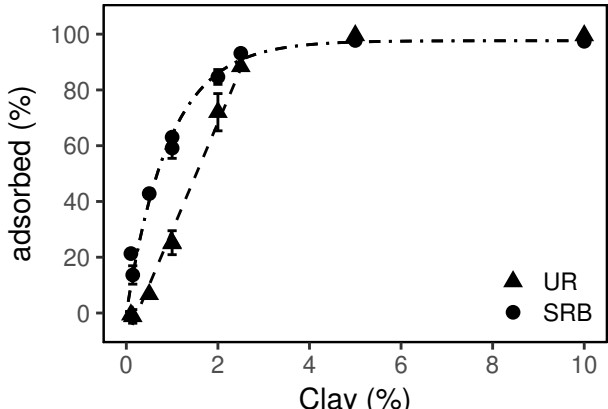

**Figure 6.** Adsorbed percentage as a function of the clay content for UR (triangles) and SRB (circles) in the sediment. The data points at 0.14, 1 and 2 % clay- addition contain 18 single measurements consisting of the six tracer concentrations and each in triplicate preparation. The errorbars represent the standard deviation. The other data-points (without errorbars) are single measurements of tracer adsorption at 0.1, 0.5, 2.5, 5.0 and 10.0 % clay addition. The dashed line is the linear regression line of UR adsorption up to 2.5 % clay ($UR_{ads.} = 38.79 \cdot Clay - 9.33$, $r^2 = 0.988$). The dot-dashed line is the exponential regression line of SRB adsorption up to 10 % clay ($SRB_{ads.} = 97.65 \cdot (1 - e^{(-1.06 \cdot Clay)})$.





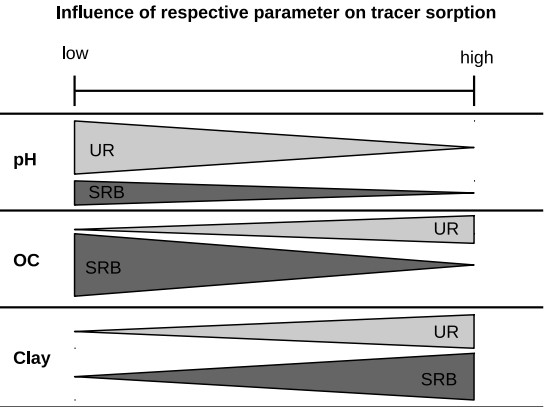

**Figure 7.** Schematic figure of the influence of the parameters OC, Clay and pH on the adsorption of the fluorescent tracers UR and SRB.




**Table 1.** $K_d$-values for UR and SRB found by several researchers.

| $K_d$ ($L \cdot kg_{soil}^{-1}$) UR | SRB | Exp.[*] | Sand | Silt (%) | Clay | pH (-) | OC (%) | $t_{batch}$ (h) | Source |
|---|---|---|---|---|---|---|---|---|---|
| 1 - 9.7 | 4.7 - 77.4 | B | 7.8 - 77 | 13 - 62 | 8 - 35 | 6.0 - 8.5 | 0.5 - 8.6 | 24 | (Dollinger et al., 2017) |
| - | 1.8 - 3.2 | C | 20.8 | 72.8 | 6.4 | 7.5 | 5.2[a] | - | (Durst et al., 2013) [b] |
| 0 | 2.9 | B | 10 | 80 | 10 | - | 5.3 | - | (Schuetz et al., 2012) |
| - | 4.4 - 9.1 | B | | silty loam | | 7.2 - 7.3 | < 4.8 | 48 | (Albrecht et al., 2003) |
| 10 | - | B | | limestone | | neutral | 0.8 | > 24 | (Sabatini, 2000) |
| - | 1.2 - 3.2 | B | | sand stone | | neutral | 0.04 | > 24 | (Sabatini, 2000) |
| - | 2.4 - 18.3 | B | 45 | 34 | 21 | 3.6 - 4.0 | 2.0 - 0.3 | 48 | (Vanderborght et al., 2002) |
| 0.05 - 0.33 | - | B/C | 97.3 | 2.2 | 0.5 | 7.9 | 0.3 | 2 | (Sabatini and Austin, 1991) |
| 0.28 - 0.4 | 29.4 - 36.6 | B | 79 | 19 | 2.0 | - | - | - | (Mägdefessel, 1990) [c] |
| 0.48 | 42 | B | 79 | 19 | 2 | - | - | - | (Mägdefessel, 1990) [d] |
| 0 | - | B | | sandy material | | prob. > 7.0 | - | - | (Dervey, 1985; Wernli, 1986) [c] |
| 0 | 4.8 - 8.6 | B | | sandy material | | - | - | - | (Wernli, 1986) [d] |
| 1.3 - 10.3 | - | B/C | | loamy sand | | 6.5 - 7.0 | 2 | 24 | (Omoti and Wild, 1979a) |
| 0.2 - 4.8 | 1.3 - 15.3 | B | 97.6 - 95.6 | 2.3 | 0.1 - 2.1 | 7.5 | 0.8 | 48 | this study |
| 0.6 - 73.2 | 2.7 - 19.3 | B | 25.4 - 26.3 | 50.6 - 51.3 | 22.4 - 24.0 | 5.5 - 7.5 | 0.1 - 2.7 | 48 | this study |

[*] *B: Batch experiment; C: Column experiment*

[a] *Total organic carbon*

[b] *Boundaries of sand and silt size ranges differ slightly in this study: Sand [50-1000 μm], Silt [2-50 μm].*

[c] *cited in Leibundgut et al. (2009)*

[d] *cited in Wernli (2011)*



**Table 2.** Texture, pH-values, OC-, and metal oxide-contents (mean of three measurements), DOC concentrations (median of two measurements) and specific surface area (median of two measurements) of top- and subsoil and the sediment in dependence of the respective treatment (high or low OC). The treatment "high OC" are the untreated samples and "low OC" the $H_2O_2$-treated samples. $Me_o$ is the summed content of all oxalate extractable metal oxides ($Fe_o + Al_o + Mn_o$) and $Me_d$ is the summed content of all dithionite extractable metal oxides ($Fe_d + Al_d + Mn_d$).

| Substrate | Treatment | OC (%) | Sand : Silt : Clay (%) | pH$^g$ (-) | $Me_o$ $^d$ (mmol · kg$^{-1}$) | $Me_d$ $^d$ (mmol · kg$^{-1}$) | $Me_o$ / $Me_d$ - | DOC $^e$ (mg · L$^{-1}$) | SSA $^e$ (m$^2$ · g$^{-1}$) |
|---|---|---|---|---|---|---|---|---|---|
| Topsoil | low OC | 0.7 | 25.4 : 50.6 : 24.0 | 4.8 / 4.8 | 137 ± 4.0 | 206 ± 4.1 | 0.67 | 45 ± 0.3 | 12 ± 0.02 |
| Topsoil | high OC | 2.7 | | | | | | 29 ± 0.03 | 10 ± 0.12 |
| Subsoil | low OC | 0.1 | 26.3 : 51.3 : 22.4 | 4.3 / 4.3 | 92 ± 2.6 | 243 ± 12 | 0.38 | 42 ± 0.9 | 24 ± 0.03 |
| Subsoil | high OC | 0.6 | | | | | | 10 ± 1.4 | 24 ± 0.31 |
| Clay (pure)$^a$ | - | - | 0 : 0 : 100 | - | - | - | - | - | 249 ± 0.28$^f$ |
| Sediment | Clay0 | 0.8 | 97.6 : 2.3 : 0.1 | 9.1 / 8.1 | 2 ± 0.04 | 18 ± 2.4 | 0.10 | 4 ± 0.1 | 1.1 ± 0.04 |
| Se+Clay$^b$ | Clay1 | - | 96.6 : 2.3 : 1.1$^c$ | - | - | - | - | - | 3.6 $^c$ |
| Se+Clay$^b$ | Clay2 | - | 95.6 : 2.3 : 2.1$^c$ | - | - | - | - | - | 6.1 $^c$ |

$^a$ Montmorillonite.

$^b$ Addition of 1 % or 2 % of the clay mineral montmorillonite.

$^c$ Values are calculated from Clay content and SSA for sediment and the pure clay mineral.

$^d$ Mean and standard deviation of three measurements.

$^e$ Range of two measurements.

$^f$ Standard deviation of four measurements.

$^g$ pH in $H_2O$ / pH in 0.01 M $CaCl_2$-solution.





**Table 3.** Characterisation of fluorescent tracers.

| Parameter | Uranine | Sulforhodamine B |
|---|---|---|
| Molecular Structure | | |
| Cas-No. | 518-47-8 | 3520-42-1 |
| C.I. (Name) | Acid Yellow 73 [a] | Acid Red 52 [a] |
| Chem. formula | $C_{20}H_{10}O_5Na_2$ | $C_{27}H_{29}N_2NaO_7S_2$ |
| Toxicity | Harmless [a] | Sufficient [a] |
| $M_W$ (g $\cdot$ mol$^{-1}$) | 376 | 580 |
| Ex. / Em. (nm) | 491/516 [a] | 560 / 585 [a] |
| Solubility (g $\cdot$ L$^{-1}$) | 300 (20 °C) [a] | 10 (10 °C) [a] |
| Relative fluorescence intensity (%) [*] | 100 [a] | 7 [a] |
| $pK_a$ [-] | 1.95 / 5.05 / 7.00 [b] | < 1.5 [c] |
| $\log(K_{OW})$ | 3.35 [d] / -0.67 [e] | 1.3 [f] |

[a] *Source:Leibundgut et al. (2009)*

[b] *Source: Zanker and Peter (1958)*

[c] *Source: Kasnavia et al. (1999)*

[d] $\log(K_{OW})$ for the twofold protonated (neutral) species (dominant species between pH 1.95 and 5.05), Calculated with *U.S. Environmental Protection Agency (2012)*

[e] $\log(K_{OW})$ for the disodium salt (twofold negative charged) species (dominant species above pH 7.00), Calculated with *U.S. Environmental Protection Agency (2012)*

[f] $\log(K_{OW})$ at pH 7.15, *Source: Scientific Committee on Consumer Products (2008)*

[*] values in pure natural water (groundwater)





**Table 4.** $K_d$-values for UR and SRB adsorption in topsoil, subsoil and sediment at pH 5.5, 6.5 and 7.5 with different clay and OC treatments. The OC treatment "high OC" represents the untreated samples and "low OC" the $H_2O_2$-treated samples. The clay treatments Clay0, Clay1 and Clay2 represent the sediment with 0 %, 1 % and 2 % clay additions, respectively.

| pH | Substrate | Treatment | $K_d$ [a] | conf. interval | $K_d$ [a] | conf.interval | $R^2$ | |
|---|---|---|---|---|---|---|---|---|
| (-) | | | $(L \cdot kg_{soil}^{-1})$ | | $(L \cdot kg_{soil}^{-1})$ | | (-) | |
| | | | UR | | SRB | | UR | SRB |
| 5.5 | Subsoil | high OC | 28.2 ± 2.9 *** | [20.1, 36.2] | 19.3 ± 1.2 *** | [16.1, 22.6] | 0.96 | 0.99 |
| 5.5 | Topsoil | high OC | 73.2 ± 12.8 ** | [37.7, 108.6] | 5.3 ± 0.3 *** | [4.5, 6.1] | 0.89 | 0.99 |
| 6.5 | Subsoil | high OC | 13.9 ± 2.3 ** | [7.5, 20.3] | 13.2 ± 1.1 *** | [10.3, 16.2] | 0.90 | 0.98 |
| 6.5 | Topsoil | high OC | 46.7 ± 7.7 ** | [25.3, 68.2] | 4.2 ± 0.3 *** | [3.4, 5.1] | 0.90 | 0.98 |
| 7.5 | Subsoil | low OC | 2.3 ± 0.5 ** | [1.0, 3.6] | 16.4 ± 1.4 *** | [12.4, 20.3] | 0.86 | 0.97 |
| 7.5 | Subsoil | high OC | 2.3 ± 0.2 ** | [1.6, 2.9] | 9.3 ± 0.3 *** | [8.3, 10.2] | 0.96 | 0.99 |
| 7.5 | Topsoil | low OC | 0.6 ± 0.3 * | [-0.3, 1.4] | 8.4 ± 0.6 *** | [6.8, 10.0] | 0.46 | 0.98 |
| 7.5 | Topsoil | high OC | 3.3 ± 0.4 *** | [2.2, 4.4] | 2.8 ± 0.3 ** | [1.9, 3.7] | 0.95 | 0.95 |
| 7.5 | Sediment | Clay0 | 0.2 ± 0.2 * | [-0.2, 0.7] | 1.3 ± 0.3 * | [0.4, 2.2] | 0.34 | 0.80 |
| 7.5 | Se + 1 % Clay [b] | Clay1 | 0.6 ± 0.2 * | [0.1, 1.1] | 4.5 ± 0.3 *** | [3.6, 5.3] | 0.74 | 0.98 |
| 7.5 | Se + 2 % Clay [b] | Clay2 | 4.8 ± 0.4 *** | [3.7, 5.9] | 15.3 ± 1.0 *** | [12.5, 18.2] | 0.97 | 0.98 |

[a] $K_d$-values were determined by linear regression and are presented ± standard error

[b] Addition of 1 % or 2 % of the clay mineral montmorillonite.

significance levels of $K_d$-values: (***) p < 0.001; (**) p < 0.01; (*) p < 0.05