# Peer review of "Controls of fluorescent tracer retention by soils and sediments"

_Hydrology and Earth System Sciences, 2019_

## Referee Comment (RC1) · Anonymous Referee #1 · 8 Jun 2019

General comment: The manuscript deals with batch investigations on uranine and sulforhodamine B adsorption in soils and to quantify their impact and possible interactions. In particular, the effects of several parameters, such as soil composition (clay and organic matter) and pH have been investigated. The manuscript is suitable to be published in this journal; however, some points should be addressed before publication. Some minor language mistakes are present that should anyway be corrected.

Specific comments: 2.3.2 Sorption isotherms of the tracers It is not clear the reason because you investigated the adsorption of tracers by batch tests and not by using column tests, considering the variation of humidity along the column. Please, support your approach. 3. Results and discussion Please, improve comparison between experimental findings and literature data.

---

## Referee Comment (RC2) · Anonymous Referee #2 · 14 Aug 2019

The manuscript "Controls of fluorescent tracer retention by soils and sediments" aimed at determining the use and impact of uranine (UR) and sulforhodamine B (SRB) as artificial tracers in soils and sediments.

The study is very interesting and in line with the scope of the journal but it's not clearly organised and the following amendments require to be addressed previous publication.

1- Abstract too detailed. Requirements of the journal specify the abstract to include a brief introduction of the topic, the summary recapitulates the key points of the article and mentions possible directions for prospective research. I suggest the authors to revise the abstract and insert less details on the experiments set up. 2- Pg 2- Line 12- This "multi-tracer approach") - please edit with "multi-tracer approach" 3- Pg 3 line 13-19. Please correct your English. 4- Introduction: From line 20 to 33, there are details

related to the methods used which should be moved to the related section. 5- Materials and methods- should include a section describing the case study and all characteristics on the soil and sediments analysed. 6- References to the figures and tables are vague. 7- The method used in not very well explained. The different sub-paragraphs should be better addressed and introduced. 8- Figures captions are too long. Figure 2-6. The description of the figures should be added to the text rather than having long figure titles. 9- Page 4 line 9. Tab S1- Not clear this reference as there is a Table 1.

---

## Author Comment (AC1) · 16 Sep 2019

We wish to acknowledge the constructive and helpful comments of the reviewer. The comments identified important areas that still required improvement. Below, we described point by point how we addressed the comments (in italics) in the revised paper. In blue you will find the positions of the corresponding changes in the revised manuscript:

**General comments**

**Comment 1:** *Some minor language mistakes are present that should anyway be corrected.*

[Figure]

**Response 1:** We thank the referee for this comment. An experienced colleague and native speaker checked the language once again.

**Specific comments:**

**2.3.2 Sorption isotherms of the tracers**

**Comment 2:** *It is not clear the reason because you investigated the adsorption of tracers by batch tests and not by using column tests, considering the variation of humidity along the column. Please, support your approach.*

**Response 2:** We thank the referee for this suggestion. According to the first review, we supported our approach in the introduction (p.3, l.26-30). We explained that batch tests have the main advantage over column tests that the experimental conditions can be precisely controlled. Under these experimental conditions, adsorption properties can be investigated independent of transport processes like preferential flow and transport-related soil properties like porosity etc. In the revised manuscript we have now additionally supported our approach in the discussion (p.10,l.24-26).

**Revision in the marked manuscript:** p.3, l.26-30 (Introduction); p.10, l.24-26 (Results and discussion)

**3. Results and discussion**

**Comment 3:** *Please, improve comparison between experimental findings and literature data.*

**Response 3:** We thank the referee for repeating this important suggestion. We added additional comparisons between experimental findings and literature data to the revised manuscript.

**Revision in the marked manuscript:** p.10,l.19-22; table 1; p.10/p.11,l.31/l.1 (Results and discussion)

Please also note the supplement to this comment:
https://www.hydrol-earth-syst-sci-discuss.net/hess-2019-229/hess-2019-229-AC1-supplement.pdf

---

## Author Comment (AC2) · 16 Sep 2019

We thank the reviewer for spending his time to review our manuscript. We appreciate his suggestions that definitely helped us to improve our manuscript. Below, we described point by point how we addressed the comments (in italics) in the revised paper. In blue you will find the positions of the corresponding changes in the revised manuscript:

**Comment 1:** *Abstract too detailed. Requirements of the journal specify the abstract to include a brief introduction of the topic, the summary recapitulates the key points of the article and mentions possible directions for prospective research. I suggest the authors to revise the abstract and insert less details on the experiments set up.*

[Figure]

**Response 1:** We thank the referee for this important comment. We revised the abstract, which is more tense now.

**Revision in the marked manuscript:** p.1/p.2, l.2-27/l.1-6 (Abstract)

**Comment 2:** *Pg 2- Line 12- This "multi-tracer approach") - please edit with "multi-tracer approach".*

**Response 2:** Thank you for this remark. The typing error was corrected.

**Revision in the marked manuscript:** p.2, l.18 (Introduction)

**Comment 3:** *Pg 3 line 13-19. Please correct your English.*

**Response 3:** Thank you for this comment. An experienced colleague who is a native speaker checked this paragraph once again.

**Revision in the marked manuscript:** p.3, l.18-24 (Introduction)

**Comment 4:** *Introduction: From line 20 to 33, there are details related to the methods used which should be moved to the related section.*

**Response 4:** We thank the referee for this important suggestion. We shortened the respective paragraph and removed unnecessary details.

**Revision in the marked manuscript:** p.2, l.26-32 (Introduction)

**Comment 5:** *Materials and methods- should include a section describing the case study and all characteristics on the soil and sediments analysed.*

**Response 5:** We thank the referee for encouraging us to support our materials and methods section. The characterization and analysis of soil and sediment samples is described in section 2.1 ("The substrates"). Here we describe the sample preparation (p.4, l.11-12), the determination of the residiual gravimetric water content (p.4, l.12-15), the texture (p.4, l.15-16), the pH (p.4, l.16-17), the DOC concentration (p.4, l.18-20). Furthermore we described the measurement of OM (p.4, l.22-26), the specific surface

area (p.5, l.27-29), the metal oxide contents (p.4, l.30-33/p.5, l.1) and the measurement of all element concentrations by ICP-OES (p.5, l.1-2).

**Comment 6:** *References to the figures and tables are vague.*

**Response 6:** We thank the referee for this comment. We have added additional references to figures and tables to the text, which are now better integrated into the text.

**Revision in the marked manuscript:** p.3, l.7 (Tab.1); p.6, l.3 (Fig.1); p.8, l.1 (Tab.2)

**Comment 7:** *The method used in not very well explained. The different sub-paragraphs should be better addressed and introduced.*

**Response 7:** We thank the referee for this suggestion. We have introduced and better addressed the subsections of the methods section in the revised manuscript.

**Revision in the marked manuscript:** p.4, l.7-8 (The substrates); p.5, l.18-19 (2.3.1 Substrate treatments); p.6, l.6 (3.2.2 Sorption isotherms)

**Comment 8:** *Figures captions are too long. Figure 2-6. The description of the figures should be added to the text rather than having long figure titles.*

**Response 8:** We thank the referee for this important suggestion. We distinctly shortened the figure titles and added the removed information to the material and methods section where it was still missing.

**Revision in the manuscript:** Fig.2 -> some text to p.6, l.20-22; Fig.3 -> removed information, already written in M&M: p.7, l.4; Fig.4 -> removed information, already written in M&M: p.5, l.20-21; Fig.5 -> removed information, already written in M&M: p.7, l.4; Fig.6 -> removed information, already written in R&D: p.11, l.3 and l.17

**Comment 9:** *Page 4 line 9. Tab S1- Not clear this reference as there is a Table 1.*

**Response 9:** Thanks for this remark. "Tab. S1" refers to Table S1 that can be found

in the supplement to the manuscript.

Please also note the supplement to this comment:
https://www.hydrol-earth-syst-sci-discuss.net/hess-2019-229/hess-2019-229-AC2-supplement.pdf

**Supplement:**

[revised manuscript text omitted]

[a] $K_d$-values were determined by linear regression and are presented $\pm$ standard error

[b] Addition of 1 % or 2 % of the clay mineral montmorillonite.

significance levels of $K_d$-values: (***) $p < 0.001$; (**) $p < 0.01$; (*) $p < 0.05$

---

## Referee Report (RR1)

**Controls of fluorescent tracer retention by soils and sediments**

The manuscript "Controls of fluorescent tracer retention by soils and sediments" aimed at determining the use and impact of uranine (UR) and sulforhodamine B (SRB) as artificial tracers in soils and sediments.

The study has been highly improved, although more corrections need to be addressed previous acceptance. In addition, some informal expressions are used which are not acceptable for scientific manuscripts.

1- The English is still to revise in the whole paper, as example some parts are specified in the following:

Lines 3-4 pg 1. "Recently, attempts have been made to use such dyes to trace organic pollutants in soil, but the controls of sorption of UR and SRB in soils are still incomplete and poorly standardized".
Lines 10-13 pg 2 : "Some studies suggest that it is generally possible to use fluorescent tracers to mimic the transport of organic pollutants (Sabatini and Austin, 1991; Passeport et al., 2010; Lange et al., 2011; Durst et al., 2013), although detailed insights into internal mobility controlling processes are missing so far, especially for soil systems.".. so far?
Lines 18-20 pg 2: Smart and Laidlaw (1977) investigated the sorption of several dye tracers, amongst others UR and SRB, on different organic (humus, sawdust) and inorganic (kaolinite, limestone, orthoquartzite) materials". .. amongst others?
Lines:
Lines 24-25 pg 2: "UR and SRB adsorbed stronger on positively charged surfaces than on negatively charged ones".. ones?
Lines 29-30 pg 2: "In the past, UR and SRB fluorescence tracers are mainly used in hydrogeological research to identify water flow in saturated zones such as aquifers and karst regions".. In the past.. are..?

Lines 23-24 pg 5 : "In the following, the tracer-soil suspension was agitated for 42h until sorption equilibrium".. in the following?
Line 23 pg 7 : "The adsorption of UR in top- and subsoil strongly decreased with increasing pH" should be corrected to "The adsorption of UR in top- and subsoil strongly decreased by increasing pH" and accordingly all similar sentences.

2- Please better explain the following sentences:
Lines 17-18 pg 2:"A few sorption studies of fluorescent tracers were carried out using pure organic or inorganic sorbents but the controls of sorption were examined rather by random than by systematically controlled experiments (Tab.1)".

3- In the introduction section on the paper it should be specified what types of contaminant behaviour are better described by using uranine (UR) and sulforhodamine B (SRB) dyes.

4- The authors describe both UR and SRB to be good tracers to predict the fate and transport of similar substances and analyse flow pathways, measure water velocities and determine hydrodynamic dispersion in groundwater, although their experiments are limited to consider batch tests and to obtain sorption isotherms. They should indicate why they only considered them.

---

## Referee Report (RR2)

Comment to the paper: "Controls of fluorescent tracer retention by soils and sediments" by Marcus Bork et al.

**General comment:**

The manuscript has been revised and a general improvement can be found. The paper is suitable to be published in this journal in the current form.

---

## Author Response (AR2)

**Second response to referee #2 of the review of the manuscript: Controls of fluorescent tracer retention by soils and sediments (hess-2019-229): Report from 18 Nov 2018**

Marcus Bork[1,2], Jens Lange[2], Markus Graf-Rosenfellner[1], and Friederike Lang[1]

[1]Soil Ecology, Faculty of Environment and Natural Resources, University of Freiburg, Freiburg, Germany
[2]Hydrology, Faculty of Environment and Natural Resources, University of Freiburg, Freiburg, Germany

**Correspondence:** Marcus Bork (marcus.bork@bodenkunde.uni-freiburg.de)

**1  Answers to referee #2**

We thank the referee for spending his time to review our manuscript again. We would also like to thank the editor for once again giving us the opportunity to further improve our manuscript. Below, we described point by point how we addressed the comments (in italics) in the revised paper. In blue you will find the revisions and the positions of the corresponding changes in the revised manuscript:

**Comment 1:**  *1- The English is still to revise in the whole paper, as example some parts are specified in the following:*

**Response 1:**  We thank the referee for his time and effort in listing the language mistakes. The manuscript was edited twice by Kenton Stutz (see acknowledgement), a native speaker, and we apologize for overlooking some mistakes. We did our best again to find and correct them and hope to have fixed them now. Below, after the wrong sentence in italics follows in blue the corrected sentence:

- *Lines 3-4 pg 1. "Recently, attempts have been made to use such dyes to trace organic pollutants in soil, but the controls of sorption of UR and SRB in soils are still incomplete and poorly standardized."*

- Recently, attempts have been made to trace organic pollutants in soil with such dyes, but the knowledge on the controls of sorption of UR and SRB in soils is still incomplete and poorly standardized. (p.1, l.3-4)

- *Lines 10-13 pg 2 : Some studies suggest that it is generally possible to use fluorescent tracers to mimic the transport of organic pollutants (Sabatini and Austin, 1991; Passeport et al., 2010; Lange et al., 2011; Durst et al., 2013), although detailed insights into internal mobility controlling processes are missing so far, especially for soil systems..." so far?*

- Some studies suggest that it might generally be possible to use fluorescent tracers to mimic the transport of organic pollutants (Sabatini and Austin, 1991; Passeport et al., 2010; Lange et al., 2011; Durst et al., 2013), although detailed knowledge on mo- bility controlling processes, especially for soil systems, is largely missing. (p.2, l.14-17)

- *Lines 18-20 pg 2: Smart and Laidlaw (1977) investigated the sorption of several dye tracers, amongst others UR and SRB, on different organic (humus, sawdust) and inorganic (kaolinite, limestone, orthoquartzite) materials". .. amongst others?*

- Smart and Laidlaw (1977) investigated the sorption of several dye tracers, including UR and SRB, on different organic (humus, sawdust) and inorganic (kaolinite, limestone, orthoquartzite) materials. (p.2, l.24-26)

- *Lines 24-25 pg 2: "UR and SRB adsorbed stronger on positively charged surfaces than on negatively charged ones".. ones?*

- UR and SRB showed stronger adsorption on positively than on negatively charged surfaces. (p.2, l.30-31)

- *Lines 29-30 pg 2: "In the past, UR and SRB fluorescence tracers are mainly used in hydrogeological research to identify water flow in saturated zones such as aquifers and karst regions".. In the past.. are..?*

- This sentence has been completely changed: As described above, UR and SRB have mainly been used in systems where sorption plays a minor role. (p.3, l.1-4)

- *Lines 23-24 pg 5 : "In the following, the tracer-soil suspension was agitated for 42h until sorption equilibrium".. in the following?*

- Then, the tracer-soil suspension was agitated for 42h until sorption equilibrium. (p.6, l.3-4)

- *Line 23 pg 7 : "The adsorption of UR in top- and subsoil strongly decreased with increasing pH" should be corrected to "The adsorption of UR in top- and subsoil strongly decreased by increasing pH" and accordingly all similar sentences.*

- Here we consulted our native speaker again: "increase by" would be followed by a rate or number, "increase with" means a tendency of relationship. So "with" is correct in the way we use it here.

- Further language corrections are made directly in the manuscript and are not listed here again.

**Comment 2:** *2- Please better explain the following sentences: Lines 17-18 pg 2:"A few sorption studies of fluorescent tracers were carried out using pure organic or inorganic sorbents but the controls of sorption were examined rather by random than by systematically controlled experiments (Tab.1)".*

**Response 2:** We thank the referee for this comment as it shows that this point has not been made clear enough. This sentence is explained in detail in the paragraph (p.2, l. 22-34) that is connected to this sentence. Here, we explain that, for example, Smart and Laidlaw (1977) investigated the sorption of UR and SRB on organic and inorganic materials but did not control the pH. Therefore, it remains unclear whether the differences in adsorption of tracers are related to the pH or to the content of organic matter. This is what we mean by "systematically controlled". All parameters except one are fixed. Hence, differences in adsorption can be clearly attributed to this one parameter. We modified the sentence to make this point more clear:

**Revision in the marked manuscript:** A few sorption studies of fluorescent tracers were carried out using pure organic or inorganic sorbents. However, the controls of sorption were not examined by systematically controlled experiments (Tab.1) that varied only one parameter and left all others constant. (p.2, l.22-24)

**Comment 3:** *3- In the introduction section on the paper it should be specified what types of contaminant behaviour are better described by using uranine (UR) and sulforhodamine B (SRB) dyes.*

**Response 3:** We thank the referee for this comment. We included a corresponding sentence in the introduction.

**Revision in the marked manuscript:** So far, UR was mainly used in surface waters as a tracer for light decay (Lange et al., 2011, Gutowski et al., 2015) while SRB was found to be suitable to mimic adsorption on sediments (Durst et al., 2013, Dollinger et al., 2017). (p.2, l.6-8)

**Comment 4:** *4- The authors describe both UR and SRB to be good tracers to predict the fate and transport of similar substances and analyse flow pathways, measure water velocities and determine hydrodynamic dispersion in groundwater, although their experiments are limited to consider batch tests and to obtain sorption isotherms. They should indicate why they only considered them.*

**Response 4:** The statement that UR and SRB are used to predict the fate and transport of similar substances is deduced from the literature (p.1, l.25-26) from studies that used them in groundwater, i.e systems where sorption plays a limited role. The aim of our study was to investigate the sorption properties of UR and SRB in soils and sediments, i.e. systems where sorption is important. Batch tests are suitable tools for this purpose as individual factors that influence sorption can be precisely controlled. We have revised the corresponding paragraph in the introduction and hope it is clearer now.

**Revision in the marked manuscript:** As described above, UR and SRB have mainly been used in systems where sorption plays a minor role. However, in order to use these tracers in soils and sediments, i.e. systems where sorption is the main process that controls solute retention, their sorption properties must be very well known. For this reason, the present study aims to identify the main controls of UR and SRB adsorption in soils and sediments. In particular, we investigated the influence of clay, OM, pH and their interactions on tracer adsorption by selectively controlling each factor in batch experiments. These types of experiments have the main advantage that experimental conditions can exactly be defined. Batch experiments are appropriate tools to investigate the influence of a single factor independent of transport processes (i.e. preferential flow) or transport-related soil properties (i.e. porosity). To face the challenge of experiments with complex matrices, such as soils and sediments, we did not only compare natural substrates but also manipulated them in a controlled way. (p.3, l.1-20)

**Comment 5:** *5- The introduction to a research paper simply introduces the topic being researched. It should be relatively brief, concise and clear. The introduction contains the aim and objectives of the paper and then three to five reasons, details and/or facts supporting your research followed by a conclusion. The final part of the Introduction of the manuscript should be revised as it contains details should be added to Material and methods and not conclusions were presented.*

**Response 5:** We agree with the referee and thank him for raising this point. We removed the corresponding paragraph from the introduction (p.3, l.17-20) and inserted the references that justify our experimental treatments in the material and methods section (p.5, l.10, l.15, and l.22).

**Comment 6:** *6- Please indicate a reference justifying your omissions in the experiment set up and assumptions.*

**Response 6:** We thank the referee for this comment. As the referee recommended, the justification references for our treatments are now included in the material and methods section (p.5, l.10, l.15, and l.22). For more clarity, we changed a sentence within the material and methods section and justified it with a further reference (p.5, l.32, p.6, l.1). Additionally, we included a further reference in the data analysis section (OECD, 2000) (p.6, l.17).

**Revision in the marked manuscript:** The tracer concentrations were low and still in the range of linear sorption. In this case, sorption sites are in excess and far from being saturated (Schwarzenbach et al., 2003). Therefore, we assumed that the tracers did not compete during sorption. (p.5, l.27-29)

**Comment 7:** *7- As specified by the authors, linear sorption isotherms were considered as "it is widely used and enables comparison with literature data". Please better explain why this was not an outcome from experiments and results.*

**Response 7:** We thank the referee for this comment. As explained in the material and methods section (p.6, l.8-11), we identified the linear sorption range in pre-tests. The main experiment was then conducted with these concentrations. We revised the corresponding sentence and hope that it is clearer now.

**Revision in the marked manuscript:** According to our pre-tests, these tracer concentrations were in the linear sorption range and were high enough to obtain a clear tracer signal to background ratio. (p.6, l.9-11)

**Comment 8:** *8- In the result section, Figures and Tables should be clearly introduces and described in the text rather than only reported into brackets, for example Figure 1 shows results for .. etc*

**Response 8:** We thank the referee for raising this point. However, to save space and to avoid redundancies, we only cited the figures and tables in parenthesis.

For a better understanding we have added another sentence to the caption of Fig.7: 
[revised manuscript text omitted]